# Tiny miRNAs Play a Big Role in the Treatment of Breast Cancer Metastasis

**DOI:** 10.3390/cancers13020337

**Published:** 2021-01-18

**Authors:** Andrea York Tiang Teo, Xiaoqiang Xiang, Minh TN Le, Andrea Li-Ann Wong, Qi Zeng, Lingzhi Wang, Boon-Cher Goh

**Affiliations:** 1Department of Medicine, Yong Loo Lin School of Medicine, National University of Singapore, Singapore 117600, Singapore; andrea.teo@u.nus.edu (A.Y.T.T.); andrea_la_wong@nuhs.edu.sg (A.L.-A.W.); 2Cancer Science Institute of Singapore, National University of Singapore, Singapore 117599, Singapore; 3Department of Clinical Pharmacy, School of Pharmacy, Fudan University, Shanghai 20203, China; xiangxq@fudan.edu.cn; 4Institute for Digital Medicine and Department of Pharmacology, Yong Loo Lin School of Medicine, National University of Singapore, Singapore 117600, Singapore; phcltnm@nus.edu.sg; 5Department of Haematology–Oncology, National University Cancer Institute, Singapore 119228, Singapore; 6Institute of Molecular and Cell Biology, Agency for Science, Technology and Research (A*STAR), Singapore 138673, Singapore; mcbzengq@imcb.a-star.edu.sg

**Keywords:** microRNAs, breast cancer metastasis, breast cancer therapy, microRNA-based therapy

## Abstract

**Simple Summary:**

MicroRNAs (miRNAs) have emerged as important regulators of tumour progression and metastasis in breast cancer. Through a review of multiple studies, this paper has identified the key regulatory roles of oncogenic miRNAs in breast cancer metastasis including the potentiation of angiogenesis, epithelial-mesenchymal transition, the Warburg effect, and the tumour microenvironment. Several approaches have been studied for selective targeting of breast tumours by miRNAs, ranging from delivery systems such as extracellular vesicles and liposomes to the use of prodrugs and functionally modified vehicle-free miRNAs. While promising, these miRNA-based therapies face challenges including toxicity and immunogenicity, and greater research on their safety profiles must be performed before progressing to clinical trials.

**Abstract:**

Distant organ metastases accounts for the majority of breast cancer deaths. Given the prevalence of breast cancer in women, it is imperative to understand the underlying mechanisms of its metastatic progression and identify potential targets for therapy. Since their discovery in 1993, microRNAs (miRNAs) have emerged as important regulators of tumour progression and metastasis in various cancers, playing either oncogenic or tumour suppressor roles. In the following review, we discuss the roles of miRNAs that potentiate four key areas of breast cancer metastasis—angiogenesis, epithelial-mesenchymal transition, the Warburg effect and the tumour microenvironment. We then evaluate the recent developments in miRNA-based therapies in breast cancer, which have shown substantial promise in controlling tumour progression and metastasis. Yet, certain challenges must be overcome before these strategies can be implemented in clinical trials.

## 1. Introduction

Breast cancer is the most commonly occurring cancer in women and the second-most prevalent cancer overall [1], with the development of distant organ metastases accounting for 90% of breast cancer deaths [2]. Metastasis is a multi-step process characterized by (i) tumour cell invasion into adjacent tissue, (ii) transendothelial migration of cancer cells into vessels, (iii) survival in the circulatory system, (iv) extravasation to secondary tissue, and (v) subsequent proliferation and colonization in competent organs [3]. Furthermore, the Warburg effect has been shown to facilitate metastatic dissemination by minimizing oxidative stress [4], while the tumour microenvironment induces tumour growth and metastasis via various mechanisms [5].

In the past decade, microRNAs (miRNAs) have emerged as important regulators of various steps in tumour progression and metastasis. miRNAs are small non-coding RNA molecules of 19 to 24 nucleotides, which regulate gene expression in a sequence-specific fashion [6]. Following incorporation into the ribonucleoprotein RNA-induced silencing complex (RISC) via association with Argonaute 2, miRNAs often base-pair with the 3′UTRs of target mRNAs, thereby reducing mRNA translation or causing degradation of the mRNA transcript. Alternatively, miRNAs may also bind to the 5′UTR and coding sequence of the mRNA transcript [7,8]. The specific fate of the mRNA depends on the degree of base-pairing complementarity between the miRNA and mRNA [9]. This imperfect match means each miRNA possesses the potential to target multiple different mRNAs [10,11]. Furthermore, crosstalk between miRNAs and long non-coding RNAs (lncRNAs) has also been documented, with these interactions forming complex networks in targeted gene regulation [12]. Yoon et al. described four key interactions between lncRNAs and miRNAs, in which (i) miRNAs may trigger lncRNA decay, (ii) lncRNAs act as miRNA decoys, (iii) lncRNAs and miRNAs compete for target mRNAs, and (iv) lncRNAs generate miRNAs [13]. Consequently, various studies have established that miRNAs elicit either oncogenic or tumour suppressive functions by silencing target protein-coding genes [14,15], indicating the potential for miRNAs to regulate multiple signalling processes necessary for breast cancer progression and metastasis. Thus, the use of miRNAs in breast cancer therapy holds huge potential.

This review will first study the various pathways through which oncogenic miRNAs potentiate breast cancer metastasis (Figure 1), and thereafter discuss the recent developments and challenges in novel miRNA-based therapies for breast cancer.

## 2. miRNAs in Breast Cancer Metastasis

For malignant cells to metastasise, several key processes must occur (Figure 2). The induction of angiogenesis enables tumour cells to gain access to the vasculature for subsequent metastatic spread to other tissues [16]. Greater vascular density within the tumour increases the chances of tumour cell escape and intravasation [17], while the leaky and fragmented basement membranes of newly formed capillaries increase the ease of tumour cell penetrability and migration [18]. Meanwhile, the neoplastic cells may invade the surrounding stroma via collective migration or epithelial-mesenchymal transition (EMT) [19]. During invasion, the tumour cells disrupt the basement membrane and penetrate the underlying stroma, a process in which regulation of adhesion, ECM reorganization and motility must occur [20,21]. In EMT, tumour cells lose their epithelial properties and gain migratory and invasive traits. Importantly, the loss of epithelial marker E-cadherin from the adherens junctions, along with a change in the upregulation of keratin expression to the mesenchymal intermediate filament vimentin expression, is characteristic of cancer EMT [22]. It is postulated that this switch to a mesenchymal phenotype endows the cells with migratory and stem-like properties as well as reduced cell-cell adhesion [19,23].

Interestingly, the metastatic process does not occur due to the properties of the tumour cells alone. The tumour microenvironment, consisting of a variety of resident and infiltrating host cells, secreted factors and ECM proteins, plays a large role in determining the fate of the cancer cells [24]. It has been implicated in tumorigenesis, tumour progression and metastasis formation. For instance, certain cells such as cancer-associated fibroblasts and M2-type tumour-associated macrophages have been shown to assist angiogenesis [25] and tumour cell invasion, migration and eventual intravasation [26,27], while M1 macrophages have been described as anti-tumour effectors [28]. Furthermore, the unique cancer cell metabolic phenotype, known as the Warburg effect, has been shown to be a significant contributory factor in these processes as well. By constraining the pyruvate flux into mitochondrial oxidative metabolism, the Warburg effect minimizes oxidative stress from mitochondrial respiration within cancer cells. This then facilitates metastatic dissemination by providing cancer cells with a survival advantage [4,29].

Following intravasation and survival in the circulatory system, the tumour cells then extravasate to pre-metastatic niches in secondary sites that are suitable for proliferation and colonization [30]. If colonization is successful, this will result in the formation of a distant metastasis.

miRNAs have been implicated in each stage of cancer metastasis, acting either as tumour suppressors or oncogenic miRNAs to suppress or promote metastasis respectively. In some cases, miRNAs may act as both tumour suppressors and oncogenic miRNAs depending on the type of cancer and cellular context [31]. While most miRNAs are found within the cellular microenvironment, miRNAs have also been detected in the extracellular environment, protected and carried by vesicles such as exosomes [32], or associated with proteins [33]. These extracellular miRNAs function as chemical messengers to mediate cell-cell communication. For instance, exosomal release of various miRNAs by breast cancer cells has been suggested to promote their own survival and invasion, thereby promoting metastasis [34,35]. In addition, the release of miRNAs from multiple cells types could further enhance the metastatic progression of cancer through modulating the tumour microenvironment, as discussed below in Section 2.4. Indeed, multiple stages in the metastatic cascade are tightly regulated by miRNAs. The following subsections will provide a closer look at miRNAs involved in potentiating four key processes of breast cancer metastasis—angiogenesis, EMT and invasion, the Warburg effect and the tumour microenvironment.

### 2.1. miRNAs in Angiogenesis

Besides its pivotal role in tumour survival and growth, angiogenesis enables tumour cells to break off from the primary tumour and travel to distant sites via the vasculature [36]. Various growth factors and proteins secreted by tumours promote angiogenesis, including VEGF, angiopoietin 1 and 2 and TGF-ß. Notably, hypoxia induces the expression of VEGF and its receptor via HIF1α, and VEGF in turn causes ECM remodelling [37] and proliferation of blood vessels [38]. Angiopoietin 1 mediates vessel maturation, migration and adhesion of endothelial cells, while angiopoietin 2 promotes neovascularisation in conjunction with VEGF [39]. Oncogenic miRNAs have been found to stimulate this angiogenic process by regulating the expression of growth factors and proteins in angiogenesis, thereby promoting breast cancer metastasis (Table 1).

For instance, miR-155 was demonstrated to induce angiogenesis in breast cancer via its target, von Hippel-Lindau, a ubiquitin ligase that targets HIF1α [45]. Notably, extensive angiogenesis, proliferation, tumour necrosis and recruitment of pro-inflammatory cells such as tumour-associated macrophages were observed following mammary fat pad xenotransplantation of miR-155 [41]. In addition, miR-93 was found to promote tumour angiogenesis and metastasis in breast cancer by suppressing tumour suppressor LATS2 [46] expression, with increased lung metastasis demonstrated in a mouse model [43].

VEGF is also a crucial target protein of miRNA regulation in angiogenesis. miR-9-mediated E-cadherin downregulation has been found to play a role in upregulating the expression of the gene encoding VEGF via activation of ß-catenin signalling [40]. miR-20a was also found to induce angiogenic effects in breast cancer cell lines, with its expression associated with increases in mean vessel size, VEGFA expression and the presence of glomeruloid microvascular proliferations [42]. Lastly, miR-21, which regulates multiple pathways in cancer metastasis, also plays a role in tumour angiogenesis in breast cancer. Knockdown of miR-21 demonstrated suppressed tumour growth and angiogenesis by targeting the VEGF/VEGFR2/HIF1α axis in a VEGFR2-luc mouse model of breast tumorigenesis [44].

### 2.2. miRNAs in Epithelial-Mesenchymal Transition, Invasion and Migration

EMT is a crucial contributary factor in cancer metastasis, in which cells lose their epithelial properties and acquire a morphology appropriate for invasion and migration [47,48]. Mesenchymal-to-epithelial transition (MET), the reverse process of EMT, is associated with metastatic colonisation in a distant site [49,50]. Epithelial markers include E-cadherin, cytokeratin and claudin-1 while mesenchymal markers include fibronectin, N-cadherin, SNAIL, SLUG, ZEB1, TWIST and vimentin [51,52]. The loss of E-cadherin expression, which induces the formation of cell-cell contact and adherens junctions, is heavily involved in EMT [53]. Repression of E-cadherin expression is mediated by multiple transcription factors such as SNAIL, ZEB 1/2 and SLUG via changes in several pathways, including TGF-β, Wnt and NOTCH [48,54,55]. Furthermore, recent studies have highlighted a link between EMT and cancer stem cells [56,57], a small subpopulation of tumour cells with self-renewal, differentiation and tumorigenicity properties when transplanted into another animal host [58]. EMT endows stem-like properties to cancer cells, while cancer stem cells commonly exhibit EMT properties [59]. Understanding the miRNAs which tightly regulate the EMT/MET pathway in cancer metastasis is thus ideal in developing targets for breast cancer therapy.

Multiple miRNAs have been identified to participate in EMT induction (Table 2).

LZTFL1 acts as a tumour suppressor which regulates β-catenin signalling in a number of cancers [79,80], consequently activating EMT. Down-regulation of miR-21 was found to inhibit EMT-mediated metastasis of breast cancer in vitro and in vivo by promoting LZTFL1 expression via the miR-21/LZTFL1/β-catenin axis [60]. Interestingly, miR-21 was also found to regulate β-catenin signalling via activation of the Akt/β-catenin pathway through PTEN, inducing subsequent EMT [60]. Besides its role in angiogenesis, miR-9 has also been linked to the promotion of EMT. miR-9 directly supresses CDH1 and thus E-cadherin, thereby leading to increased cell motility and invasiveness [40]. This was supported by another study showing that miR-9 expression in breast tumours is associated with E-cadherin loss and vimentin expression, thus playing a probable role in EMT in breast cancer [81]. Another key miRNA shown to augment migration, invasion and metastasis in breast cancer is miR-10b. An early study indicated that TWIST-induced miR-10b expression represses homeobox D10 mRNA translation, thereby increasing pro-metastatic RHoC gene expression with tumour invasion and metastasis [61]. This was supported by subsequent studies demonstrating development of breast cancer brain metastasis with upregulation of miR-10b [82] and EMT induction in breast cancer by miR-10b, which was found to act as a target gene of TGF-ß1 [83]. It was further shown that transfection of non-malignant mammary gland epithelial cells with exosomal-derived miR-10b induced cell invasion [35].

In addition, miR-221/222 has been linked to the aggressive basal-like subtype of breast cancer through its activation of EMT [84]. miR-221/222 acts downstream of the oncogenic Ras-Raf-MEK-ERK pathway, increasing ZEB2 levels and consequently repressing E-cadherin by targeting the 3’UTR of the GATA family transcriptional repressor TRPS1, resulting in heightened EMT in basal-like breast cancer [62]. miR-374a was also found to aid in the development of a pro-metastatic phenotype of breast cancer cells in vitro via induction of EMT, with overexpression of miR-374a resulting in cell morphologies characteristic of EMT. In miR-374a-transduced breast cancer cell lines, epithelial markers including E-cadherin, γ-catenin, and CK18 were drastically downregulated, while mesenchymal markers such as vimentin and N-cadherin were upregulated. Furthermore, miR-374a was found to interfere with EMT via Wnt/β-catenin signalling by directly suppressing WIF1, PTEN, and WNT5A expression in breast cancer cell lines [63]. Hypoxia-induced upregulation of miR-191 was also demonstrated to enhance breast cancer cell proliferation, migration and survival by increasing levels of TGFß2 and downstream proteins including VEGFA both directly and indirectly [64].

However, the roles of miR-125b, miR-155 and the miR-200 family in breast cancer invasion and EMT are less clearly delineated. The tumour suppressor gene STARD13 was identified as a target protein of miR-125b. Repression of STARD13 by miR-125b in MCF-7 and MDA-MB-231 cells was responsible for EMT and metastasis in breast cancer cell lines via upregulation of vimentin and α-smooth muscle actin [67]. On the contrary, miR-125b has been found to target SNAIL-1, with knockdown of miR-125b and consequent overexpression of SNAIL-1 increasing migration, invasion, and EMT in SKBR3-TR and BT474-TR cells [65]. miR-125b was further found to reverse motility, invasion and EMT in MCF-7 and SKBR3 paclitaxel-resistant breast cancer cells by targeting SEMA4C [66], an oncogenic protein demonstrated to upregulate SNAIL and SLUG [85]. This difference in observations could possibly be reconciled by the use of different breast cancer cell lines—while miR-125b expression is reported to be repressed in most breast cancer cell lines, it appears to be elevated in MDA-MB-231 cells [67,86].

Separately, miR-155 was found to target and repress C/EBPβ, thereby potentiating TGF-ß-mediated EMT [70]. It was further discovered that mutant p53-mediated upregulation of miR-155 and silencing of its target gene ZNF652, which directly represses key drivers of invasion and metastasis, drives local invasion of breast cancers [71]. However, seemingly contradictory reports have been made. miR-155 was reported to inhibit lung metastasis from mammary fat pads by preventing EMT through suppression of TCF4 expression [68]. More recently, miR-155 was also shown to downregulate ZEB2, with consequent reduced expression of vimentin and reduced invasion. Yet, ZEB2 repression did not change E-cadherin levels, and migration was instead enhanced [69]. Overall, it is suggested that miR-155 plays a larger pro-metastatic role in breast cancer—besides its roles in EMT, it has also been demonstrated to act as an oncogenic miRNA in angiogenesis and the Warburg effect as discussed in this review.

The miR-200 family has also been found to act as either tumour suppressors or oncogenic miRNAs at different junctures in the metastatic cascade. Several studies suggest that miR-200 miRNAs are downregulated and released from cancer cells during invasion as they undergo EMT, but upregulated during colonization. These miRNAs exist in two clusters: one on chromosome 1 (miR-200b, miR-200a and miR-429) and the other on chromosome 12 (miR-200c and miR-141) [87]. The miR-200 miRNAs repress invasion and EMT by targeting ZEB1 and ZEB2, repressors of the cell-cell contact protein E-cadherin [74,75,76], and regulating genes involved with cell motility and invasion [77,88]. In particular, miR-200c directly targets actin-regulatory proteins FHOD1 and PPM1F, inhibiting cancer migration and invasion through regulation of stress fibre formation and contractility [73]. Moreover, miR-200b was found to mediate many pathways including those in axonal guidance, chemokine, epithelial adherens junction and actin cytoskeleton signalling [72]. On the other hand, it has been discovered that the expression of miR-200 miRNAs is upregulated in metastases [89,90] and its levels are also elevated in the circulation of breast cancer patients with brain metastases [91]. Uptake of extracellular vesicles containing miR-200 was found to promote MET, thereby enabling the formation of adherent cellular contacts during colonization [78]. Hence the miR-200 family is a “double-edged sword” that can act as tumour suppressor at the beginning, but as an oncogenic factor at the end of the metastatic cascade.

### 2.3. miRNAs in the Warburg Effect

First discovered in the 1920s by Otto Heinrich Warburg, the Warburg effect and its implications on tumorigenesis and cancer progression have been studied extensively [92]. The Warburg effect is characterized by the preferential metabolism of glucose via aerobic glycolysis over oxidative phosphorylation in cancer cells [93]. This process supports macromolecular synthesis by providing an abundant supply of glycolysis intermediates and promotes cancer cell proliferation [94]. Emerging literature has demonstrated the involvement of miRNAs in the regulation of cellular metabolism and the Warburg effect in breast cancer (Table 3).

Various miRNAs have been found to display oncogenic effects in pathways involved in the Warburg effect. Notably, miR-155 has been found to play an integral role in regulating various pathways of aerobic glycolysis in breast cancer [101]. miR-155 represses cMyc, a master regulator of glycolysis [102], through the PIK3R1-PDK1/Akt-FOXO3a pathway. This results in the upregulation of glucose transporters and metabolic enzymes including GLUT1, HK2, PKM2 and LDHA [95]. The study corroborates earlier research showing that FOXO3a is a direct target of miR-155 [96] and deregulates cMyc [103]. In particular, HK2 has been credited as a crucial player in the Warburg effect as it is one of the chief isozymes overexpressed in tumours that promotes aerobic glycolysis [104]. In addition to the PIK3R1-PDK1/Akt-FOXO3a pathway, further studies show that miR-155 also regulates HK2 expression in breast cancer cells in two other ways. miR-155 does so by (i) promoting the activation of STAT3, which in turn facilitates the transcription of HK2, and (ii) repressing miR-143 by targeting C/EBPβ, a transcriptional activator for miR-143, resulting in post-transcriptional upregulation of HK2 [97]. These studies have lent credence to the central role played by miR-155 in the Warburg effect in breast cancer.

Another miRNA responsible for potentiating the Warburg effect in breast cancer is miR-27b. PDH is found at the crossroads of glycolysis and the citric acid cycle, thus a loss of function or reduced expression of PDH complex components has been associated with deregulation of glucose metabolism in cancer [105]. miR-27b has been found to directly reduce the expression of PDHX, a structural component of the PDH complex. This in turn suppresses oxidative glucose metabolism and facilitates tumour growth while increasing lactate production, a distinctive characteristic of Warburg metabolism [98]. miR-378* has also been found to mediate metabolic shift in breast cancer cells by inhibiting the expression of ERRγ and GABPA, partners of PGC-1ß, a transcriptional regulator of oxidative energy metabolism [106]. Downstream effects include a reduction in tricarboxylic acid cycle gene expression and an increase in lactate production and cell proliferation, with miR-378* expression demonstrating a correlation with breast cancer progression [99].

### 2.4. miRNAs in the Tumour Microenvironment

Besides modulating the nature of breast cancer cells directly, miRNAs have been found to play an important role in the tumour microenvironment (TME). It has been suggested that the outcomes of tumour cells are largely dependent on their interaction with stromal cells; they may then either remain dormant, or progress into invasive and eventually metastatic cancer [107]. Multiple cell populations regulate the TME through intriguing mechanisms including the secretion of pro-inflammatory molecules and immune regulation (Figure 3), which will be discussed below.

Cancer-associated fibroblasts (CAFs) are one of the most common cell types found in the TME, and are responsible for the synthesis of proteins which remodel the extracellular matrix and growth factors that regulate tumour cell proliferation, survival and metastasis [108,109]. CAFs release exosomal miRNAs (miR-21, miR-378e and miR-143), increasing the stemness and EMT phenotypes of breast cancer cells [110]. miR-181d-5p was also identified in exosomes derived from CAFs, and promoted EMT via targeting CDX2, a transcription factor that binds to the homeobox A5 promoter [111]. Furthermore, miR-125b released by TNBC cells in extracellular vesicles also promotes the conversion of NFs into CAFs [112]. Interestingly, this communication between CAFs and tumour cells goes both ways: tumour-derived miRNAs have also been shown to play a role in inducing the transformation of normal fibroblasts (NFs) into CAFs [113]. For instance, exosome-mediated delivery of miR-9 from breast cancer cells induces CAF-like properties in human breast fibroblasts through modulating the expression of various extracellular matrix proteins [114]. In addition, breast cancer cell-secreted miR-205 contributes to the conversion of breast NFs into CAFs by promoting YAP1 expression and subsequent tumour angiogenesis [115], and activates Myc signalling in CAFs to induce an optimal metabolic environment for sustained tumour growth [116].

Furthermore, changes in immune cells within the tumour microenvironment may lead to inhibition of the antitumour immune response and thus breast tumour progression, with miRNAs being crucial mediators of this process. For instance, miR-375, released by breast cancer cells during apoptosis, was found to accumulate in tumour-associated macrophages and enhance phagocyte migration and infiltration in vitro and in vivo, forming a tumour-promoting microenvironment [117]. MDSCs, shown to inhibit anti-tumour T cells, were activated by doxorubicin treatment and subsequently increased breast tumour angiogenesis and induced Th2 cell activation via exosomal release of miR-126a [118]. Prostaglandin E2-induced miR-10a production has also been shown to activate AMPK signalling, which in turn promotes the expansion and activation of MDSCs [119].

In addition, miRNAs were found to modulate the function of T cells, which are associated with cancer progression [120]. Silencing of miR-126 in a murine breast cancer model caused reduced induction and suppressive function of CD4+ FOXP3+ regulatory T cells (Tregs) through the PI3K/Akt pathway, and also endowed antitumour effects of CD8+ T cells [121]. Similarly, silencing of miR-21 regulated the PTEN/Akt pathway transduction in the expansion of CCR6+ Tregs and endowed the antitumour effects of CD8+ T cells in breast tumours [122].

## 3. miRNA-Based Therapies for Breast Cancer

With strong evidence supporting the role of miRNAs in breast cancer metastasis, much research into miRNA-based therapies has been conducted in recent years. miRNA delivery is centred around two main approaches: the delivery of (i) anti-miRNA oligonucleotides (anti-miRs) against oncogenic miRNAs, or (ii) tumour-suppressor miRNA mimetics, while other approaches to oncogenic miRNA inhibition have been explored as well (Figure 4).

However, several concerns regarding these approaches have been raised. One overarching consideration would be the off-target effects of miRNA-based therapies. For one, the downstream effects of miRNAs are multiple and varied, as a single miRNA is capable of regulating transcriptional networks involving a multitude of gene transcripts [8]. In fact, most mammalian mRNAs have been shown to be conserved targets of miRNAs [123]. Thus, changing the miRNA expression levels could see diverse downstream effects aside from the intended outcome, making it challenging to avoid off-target effects [124]. Furthermore, present research on miRNA therapeutics utilising tumour suppressor miRNAs relies principally on synthetic miRNAs. This surfaces the concern of potential toxicity and immunogenicity due to the introduction of foreign genetic materials, and could contain artificial modifications that may affect the biochemical properties of the synthetic miRNAs [125]. Similarly, the delivery vehicles used in miRNA-based therapies could also result in toxicity and immunogenicity [126]. To circumvent these limitations in miRNA delivery, multiple delivery platforms have been studied with the aim of achieving a desirable balance between efficacious miRNA delivery and reduced vehicular toxicity. In the following sections, we explore the various approaches taken in miRNA-based therapies and the associated challenges for each approach.

### 3.1. miRNA Delivery

#### 3.1.1. Liposomes

Liposomes synthesized in the nanometer-size range (<250 nm in diameter) have been increasingly utilised in cancer drug delivery, and are associated with improved pharmacokinetic properties [127,128]. Achieving a good balance between size, payload concentration, drug solubility, protection from enzymatic degradation and systemic clearance has previously been emphasized in various reports [129,130]. Surface modification of liposomes can further improve the delivery of therapeutic miRNAs and antisense miRNAs against various cancers [131]. In their study, Sharma et al. developed a stearylamine based cationic liposome for the delivery of anti-miR-191 to MCF-7 and ZR-75-1 breast cancer cells. These liposomes showed efficient delivery to cancer cells with low cytotoxicity in human erythrocytes, as well as increased cancer cell apoptosis and suppressed cell migration in vitro. Furthermore, the liposomes also increased the chemosensitivity of the breast cancer cells to doxorubicin and cisplatin [132]. Recently, Lujan et al. has also described the optimization and synthesis of nanometer-sized liposomes for miRNA delivery, with miR-203 delivery to MDA-MD-231 breast cancer cells enhanced by up to 40-fold [133]. However, liposomes have limited capacity for in vivo delivery due to their rapid clearance as well as concerns over their toxicity, nonspecific uptake and immunogenicity [134].

#### 3.1.2. Inorganic Nanoparticles

Extensive research has been conducted on superparamagnetic iron oxide nanoparticles (SPIONs) for their use in biotherapeutic delivery systems [135], as they have been demonstrated to possess biocompatible and non-toxic profiles at lower therapeutic levels [136]. Separately, argonaute proteins have been found to stabilize and guide mature miRNAs to their target messenger RNAs [137,138]. By capitalizing on the special properties of SPIONs and argonaute proteins, Unal et al. designed Argonaute 2 conjugated SPIONs as tumour targeted miRNA vehicles to deliver autophagy-inhibiting miR-376b into HER2-positive breast cancer cell lines. Effective inhibition of autophagic activity by the nanoparticles was demonstrated both in vitro and in vivo in a mice xenograft model of breast cancer [139]. However, concerns over iron oxide nanoparticle-induced toxicity still remain. Toxicity of SPIONs has been shown to be dependent on exposure time and concentration: while minimal toxicity is observed at lower levels of SPIONs with good body clearance, high dose exposure to SPIONs could trigger oxidative stress and altered cellular response [140]. Thus, achieving good therapeutic efficacy with minimal toxicity is a challenge that has to be overcome before this strategy can reach clinical trials.

Using a different approach, one study reported the development of multifunctional tumour-penetrating mesoporous silica nanoparticles (MSNs) for the co-delivery of siRNA (siPlk1) and a tumour suppressor miRNA (miR-200c) to breast tumours. Previously, it was shown that iRGD, a tumour-homing and penetrating peptide, increases accumulation and penetration of anticancer drugs and nanoparticles into tumours via a three-step endocytotic transport pathway [141,142]. In addition, it has also been demonstrated that light-activated generation of ROS disrupts endosomal and lysosomal membranes, thus facilitating cytoplasmic delivery of small RNAs [143,144]. In this study, MSNs were stabilized by a surface lipid layer conjugated to iRGD. MSNs were then loaded with photosensitizer indocyanine green which generated ROS to aid endosomal escape and surface conjugation of iRGD for enhanced cytosolic RNA delivery. Upon short light irradiation, the iRGD-modified MSNs loaded with siPlk1 and miR-200c showed improved delivery, cellular uptake and tumour penetration in vitro, and significant suppression of primary tumour growth with reduction of metastasis in vivo [145]. Yet, as with most cationic nanoparticle carriers, poor elimination, immunogenicity and toxicity remain as chief concerns for MSNs [146].

Gold nanoparticles (AuNPs) have also been widely used in miRNA delivery systems. Their unique properties allow them to have low cytotoxicity, good biodistribution, tunable size and functional diversity [147]. Capitalising on these properties, Ekin et al. designed a AuNP-based nanocarrier for miR-145 transfection into prostate and breast cancer cells. Since AuNPs have a high affinity for biomolecules and can be chemically functionalised with alkyl-thiol-terminated oligonucleotides [148], the study chose to modify AuNPs with thiolated RNAs to which pre-miR-145 could then be hybridised. Effective in vitro delivery of miR-145 into MCF7 breast cancer cells was demonstrated [149]. More recently, Ramchandani et al. devised a layer-by-layer fabrication method to layer negatively charged miR-708 mimetics between positively charged PLL layers onto an inert AuNP. Subsequent degradation of the PLL layers by proteases upregulated in tumours released the miR-708 mimetics, restoring tumour suppressive miR-708 and inhibiting TNBC metastasis in vivo [150]. While promising, toxicity-related concerns over AuNPs have surfaced depending on their size [151] and ability to cross the blood-brain barrier to accumulate in neural tissue [152].

#### 3.1.3. Polymer-Based Delivery Systems

Given the associated toxicity with inorganic nanoparticles, cationic polymers have been explored as an alternative delivery platform. While concerns over the significant toxicity of first-generation carriers such as polyethylenimine (PEI) have surfaced, especially for the high-molecular weight forms [153], there is promise that this may be overcome by the introduction of biodegradable polymers in gene therapy [154]. PEIs are cationic linear or branched polymers that are able to form nanoscale complexes with small RNAs, reducing RNA degradation and increasing cellular delivery and intracellular release. They have thus been used extensively in RNA interference and gene delivery systems as an alternative to viral vectors [155]. Recent developments in PEI-based systems include disulfide cross-linked PEIs (PEI-SS) for miRNA delivery. PEI-SS was found to complex efficiently with anti-miR-155, forming nano-sized spherical structures. Subsequent biodegradation by the reducing agent glutathione in cancer cells released anti-miR-155 for the inhibition of tumour growth in vivo [156]. Another study developed a PLL-modified PEI (PEI-PLL) copolymer to transfect either miR-21 sponge plasmid DNA or anti-miR-21 oligonucleotides into MCF-7 breast cancer cells. The treated cells displayed greater miR-21 inhibition with cell cycle arrest in the G1 phase, as well as upregulation of PDCD4 involved in the caspase-3 apoptosis pathway. Furthermore, both groups of cells also showed increased sensitisation to anti-cancer drugs doxorubicin and cisplatin [157].

Poly(D,L-lactide-co-glycolide) (PLGA) is a synthetic polymer which has also been studied widely in anticancer drug delivery platforms due to its extensive functionalisation options, biodegradability, sustained-release efficacy, and stabilisation of loaded molecules [158]. Combining PLGA with PEI, Wang et al. devised a hyaluronic acid-decorated PEI-PLGA (HA/PEI-PLGA) nanoparticle system for the co-delivery of doxorubicin and miR-542-3p in TNBC therapy. Increased intracellular levels of miR-542-3p activated p53, thus promoting TNBC cell apoptosis and tumour suppression [159]. Polyethylene glycol (PEG), another biocompatible polymer, was used in combination with PGLA to form biodegradable PLGA-bPEG copolymers for the delivery of anti-miR-21 and anti-miR-10b to TNBC tumours. In vitro models displayed significant reduction in cell migration in treated cells, while in vivo models showed substantial reduction in tumour growth at low doses [160].

Another cationic polymer used in gene delivery systems is chitosan, which is biodegradable and biocompatible, and has strong nucleic acid binding affinity [161]. Several studies have established that chitosan-oligonucleotide complexes show low cytotoxicity [162,163], however, it was also reported that chitosans with higher degrees of acetylation and molecular weight may be cytotoxic [164]. In one study, chitosan was acetylated to varying degrees and used in the formation of chitosan-hsa-miR-145 (CS-miRNA) nanocomplexes, with chitosans of low degrees of acetylation forming highly stable complexes regardless of molecular weight. CS-miRNA nanocomplexes at 12% and 29% degrees of acetylation were biologically active, displaying downregulation of miR-145 target mRNA (junction adhesion molecule A mRNA) in MCF-7 breast cancer cells [165]. In addition, co-encapsulation of doxorubicin and miR-34a into hyaluronic acid-chitosan nanoparticles for simultaneous delivery into breast cancer cells was performed in another study. Enhanced anti-tumour effects of doxorubicin by suppressing the expression of non-pump resistance and anti-apoptosis proto-oncogene Bcl-2 was observed, along with the inhibition of breast cancer cell migration and metastasis by miR-34a via NOTCH-1 signalling [166].

In an unprecedented study, Conde et al. developed a self-assembling RNA-triple-helix assembly for miRNA delivery. This helix was conjugated to dendrimers and reacted with dextran aldehyde, forming an adhesive dextran-dendrimer-RNA triplex hydrogel scaffold which was able to adhere to tumour tissue and administer its miRNAs. The helix conjugate, comprising miRNA nucleotides miR-205 sense, antisense and antagomir-221, showed high structural stability and synergistic abrogation of TNBC tumours via a dual-pronged miRNA inhibition and miRNA replacement approach. Cell migration and proliferation were dramatically reduced, and nearly 90% levels of tumour shrinkage was achieved in a TNBC mouse model [167]. These results show great promise for the use of the RNA-triple-helix hydrogel in breast cancer therapy.

#### 3.1.4. Extracellular Vesicles

Another strategy to reduce vehicle-associated toxicity is the use of extracellular vesicles in gene therapy. Extracellular vesicles such as exosomes are secreted by various cell types including tumour cells, and function as natural carriers of miRNAs. These miRNAs are then taken up by recipient cells where they elicit downstream responses [168,169]. This interesting phenomenon positions extracellular vesicles as suitable miRNA delivery vehicles in breast cancer therapy due to better biocompatibility and high delivery efficiency. For instance, a tumour cell-derived extracellular vesicle (TEV)-based nanoplatform was developed for the delivery of anti-miR-21 to 4T1 breast cancer cells. Subsequent functionalisation of gold-iron oxide nanoparticles (GIONs) in TEVs to yield TEV-GIONs demonstrated the potential of TEV-GIONs for simultaneous therapy using miRNA and cancer imaging, while improving tumour-specific targeting [170]. Separately, delivery of tumour suppressor let-7a miRNA using exosomes from HEK-293 cells to EGFR-expressing xenograft breast cancer tissue in mice was performed by Ohno et al., resulting in inhibition of tumour growth and development. This was achieved by fusing a GE11 peptide, which binds specifically to EGFR, to the transmembrane domain of PDGF on donor exosomes containing let-7a. However, the same study highlighted that accumulation of exosomes was observed in the liver after injection, surfacing potential bioelimination difficulties [171].

Exosomes were also used for the co-delivery of doxorubicin, a chemotherapy drug, and hydrophobically modified miR-159 in TNBC therapy. Exosomes highly expressing ADAM15 have been found to show an enhanced binding affinity for integrin αvβ3 [172], which is overexpressed on many tumours. Hydrophobically modifying a short RNA strand via addition of a cholesterol group enables quicker membrane association and facilitates internalisation [173]. In vitro targeting of MDA-MB-231 breast cancer cells showed greater internalisation of ADAM15-rich exosomes compared to the control exosomes, and consequently significantly higher cellular uptake of doxorubicin and modified miR-159, prompting increased apoptosis. In vivo studies also showed significant tumour suppression in a xenografted-nude mouse model by ADAM15-rich exosomes [174].

One drawback of using extracellular vesicles from cell lines for miRNA or anti-miR delivery is the risk of secondary transformation. Cell lines release multiple oncogenic factors into the extracellular vesicles, thus inducing transformation of bystander benign cells in target organs. Delivery of miRNA or anti-miR oligos using extracellular vesicles from primary cells is a safer alternative, however, it is challenging to obtain sufficient extracellular vesicles from primary cells for therapeutic treatments. To overcome these limitations, a recent study by Usman et al. described the use of red blood cell-derived extracellular vesicles to deliver anti-miR-125b for an effective treatment of breast cancer and leukemia in vitro and in vivo. This approach is compelling because red blood cells are the most abundant primary cells in the body and treatment of red blood cells with calcium ionophore induces a massive release of EVs, making this an ideal approach for clinical applications [175].

### 3.2. Other Approaches to Oncogenic miRNA Inhibition

#### 3.2.1. Synthetic miRNA Sponges

First introduced in 2007, miRNA sponges contain multiple target sites complementary to a mature miRNA of interest, and are able to inhibit the activity of a family of miRNAs sharing a common seed (nucleotides 2-7 on the miRNA) [176]. The concept of miRNA sponges is a simple yet ingenious one, with researchers behind the first study reasoning that multiple binding sites could be inserted into the 3′ UTR of a decoy target to improve its affinity for its cognate miRNA. Furthermore, by designing the miRNA binding sites with a bulge at the position normally cleaved by Argonaute 2, the decoy targets would be able to stably hold on to ribonucleoprotein complexes containing numerous miRNAs [177]. Since then, multiple endogenous miRNA sponges regulating miRNA levels in breast cancer have been identified as well [178,179]. Using this concept, an miR-9 sponge construct introduced into highly malignant cells using a retroviral vector showed inhibition of metastasis formation [40]. Similar results were obtained for an miR-21 sponge, which demonstrated marked reduction in the expression of downstream CSF1, a potent activator of malignancy and metastasis [180]. More recently, Zhang et al. developed a self-assembled DNA nanosponge for the clearance of intracellular miR-21. MCF-7 breast cancer cells transfected with the nanosponge displayed increased apoptotic-related protein expression, while normal cells were minimally affected [181].

#### 3.2.2. Non-Conventional Approaches to miRNA Inhibition

Recently, novel approaches to the indirect inhibition or suppression of oncogenic miRNAs in breast cancer have been explored. In one instance, photodynamic therapy, in which ROS are produced to damage cancer cells, was used to regulate miR-155-5p expression and the Warburg effect in breast cancer. The use of 3B, a novel photosensitizer, in photodynamic therapy demonstrated impaired glucose consumption and ATP generation, inhibition of miR-155-5p expression in MCF-7 cells and decreased tumour growth in vivo [182]. In another innovative approach taken by Costales et al., a small molecule that targets the three-dimensional folds in pre-miR-21 was designed and optimised for avidity, and its target engagement of pre-miR-21 was demonstrated in MDA-MB-231 cells. Subsequent conjugation of the small molecule to a heterocyclic molecule that recruits latent ribonuclease to cleave pre-miR-21 was performed, and a drastic reduction in miR-21 levels was observed. Furthermore, this conjugated small molecule effectively inhibited invasion in multiple miR-21-expressing cancer cell lines, and inhibited breast cancer metastasis to the lung in vivo [183].

## 4. Conclusions and Future Perspectives

To conclude, miRNAs are shown to be key regulators of metastasis in breast cancer. A growing pool of studies has demonstrated the huge potential for miRNA-based therapies in breast cancer, with multiple novel approaches suggested to overcome barriers including easy degradation of RNA molecules and non-specific and off-target delivery. However, as with other nucleotide-based therapeutic approaches, concerns persist over the possibility of toxicity and immunogenicity due to the introduction of foreign genetic materials and delivery vehicles, insufficient therapeutic efficacy, off-target effects and the feasibility of upscaling production for eventual clinical applications [184]. These concerns remain particularly valid, with the phase 1 study of MRX34, a liposomal miR-34a mimic, closed early due to serious immune-mediated adverse effects that resulted in four patient deaths [185]. Furthermore, the complex regulation of multiple pathways by miRNAs may make it challenging to clearly delineate the boundary between their effects on normal versus cancer cells.

To address these concerns, several novel approaches have been developed. Instead of using synthetic miRNAs, Wang et al. suggests the use of a miRNA prodrug which can be bioengineered on a large scale in Escherichia coli, using a recombinant tRNA fusion pre-miR-34a [125]. Using a similar method, another team created an miR-127 prodrug, which was processed to mature miR-127-3p in TNBC cells and demonstrated suppressed primary tumour growth and spontaneous metastasis in vivo [186]. Separately, concerns over delivery-associated toxicity may be addressed using vehicle-free delivery systems. Orellana et al. developed a strategy to conjugate a miR-34a with folate for delivery in breast tumours [187]. Stability of the miRNA mimic was achieved by modifying the passenger miRNA strand with 2′-O-methyl RNA bases to increase nuclease resistance without impairing argonaute loading. As the folate receptor is overexpressed in breast cancers [188], this approach of miRNA delivery showed selective targeting of the breast tumour with slowing of tumour growth, offering an alternative to current methods of vehicle-based miRNA delivery. Further enhancements including sugar modifications and backbone modifications that improve cellular uptake and binding specificity of miRNAs have also been explored [189,190]. Other promising areas in RNA interference include the combination of miRNAs with chemotherapy drugs, which could show synergistic effects in breast cancer therapy [191]. Furthermore, chemical modifications to guide strand selection and delivery to reduce off-target activity could also be performed [124].

Thus, while challenges remain ahead of adoption in clinical trials, miRNA-based approaches for breast cancer therapy are definitely promising. The twin pillars of any potential miRNA-based candidates—efficacy and safety—should be rigorously validated to bring their full benefit to patients.

## Figures and Tables

**Figure 1 cancers-13-00337-f001:**
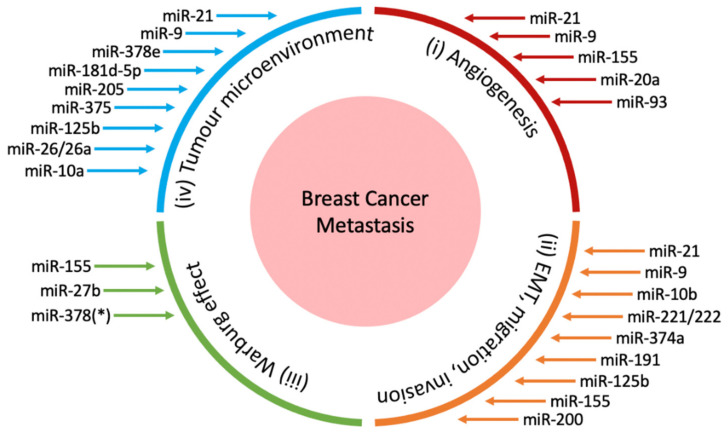
miRNAs involved in potentiating various stages and events in breast cancer metastasis: (**i**) angiogenesis, (**ii**) EMT, migration and invasion, (**iii**) the Warburg effect, and (**iv**) the tumour microenvironment.

**Figure 2 cancers-13-00337-f002:**
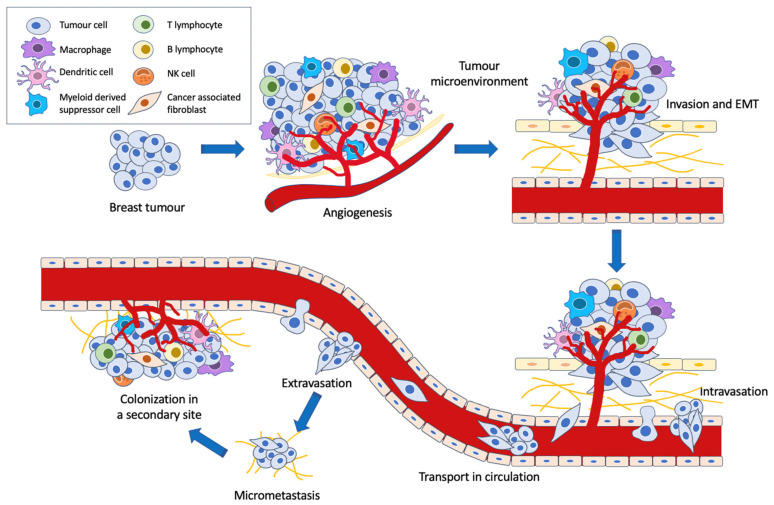
Metastatic cascade in cancer.

**Figure 3 cancers-13-00337-f003:**
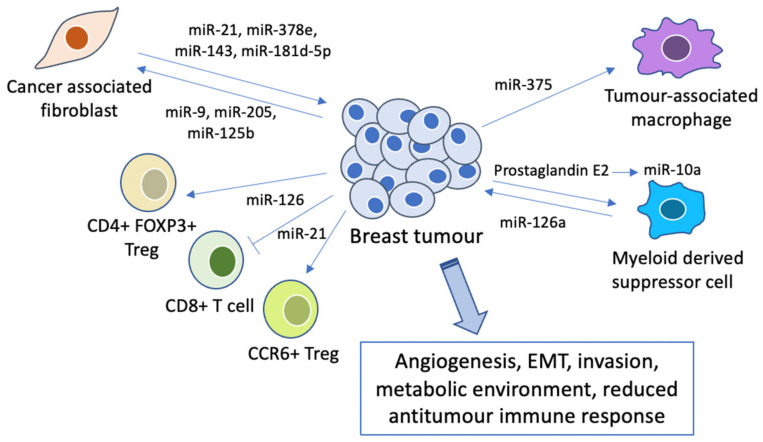
Crosstalk between breast cancer cells and stromal cells in the tumour microenvironment.

**Figure 4 cancers-13-00337-f004:**
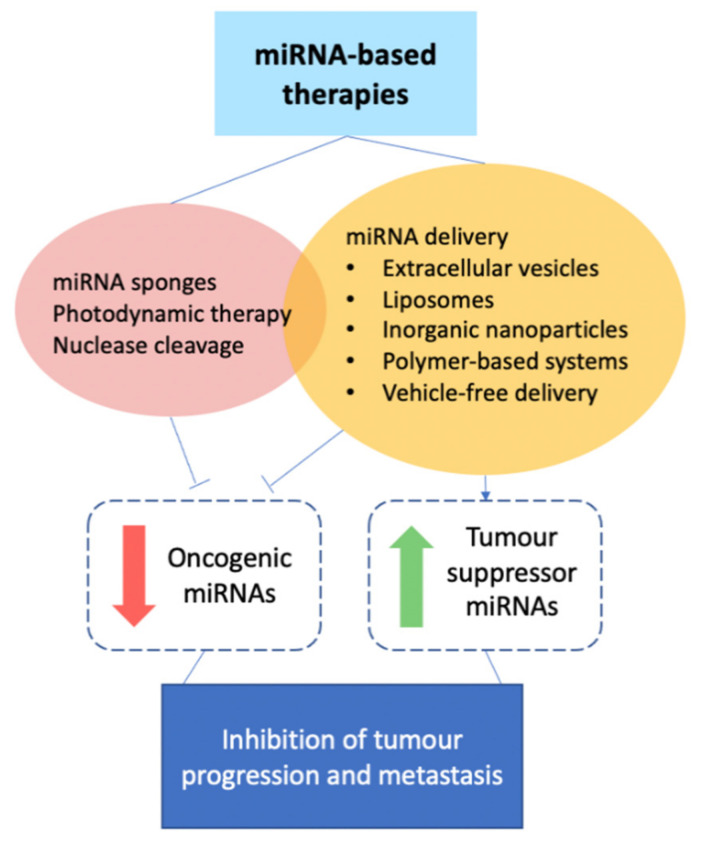
Summary of approaches taken in miRNA-based therapies for breast cancer.

**Table 1 cancers-13-00337-t001:** miRNAs involved in promoting angiogenesis.

	miRNA	Expression in Breast Cancer	Target(s)	Experimental Design	Reference(s)
Oncogenic miRNAs	miR-9	Upregulated	CDH1/ß-catenin/VEGF	In vitro & in vivo	[40]
	miR-155	Upregulated	VHL	In vitro & in vivo	[41]
	miR-20a	Upregulated	VEGFA	In vitro	[42]
	miR-93	Upregulated	LATS2	In vitro & in vivo	[43]
	miR-21	Upregulated	VEGF/VEGFR2/HIF1α	In vitro & in vivo	[44]

**Table 2 cancers-13-00337-t002:** miRNAs involved in promoting EMT, invasion and migration.

	miRNA	Expression in Breast Cancer	Target(s)	Experimental Design	Reference(s)
Oncogenic miRNAs	miR-21	Upregulated	LZTFL1, PTEN	In vitro & in vivo	[60]
	miR-9	Upregulated	CDH1	In vitro & in vivo	[40]
	miR-10b	Upregulated	Homeobox D10	In vitro & in vivo	[61]
	miR-221/222	Upregulated	TRPS1	In vitro	[62]
	miR-374a	Upregulated	WIF1, PTEN, WNT5A	In vitro & in vivo	[63]
	miR-191	Upregulated	TGFß2	In vitro	[64]
Both tumour suppressors & oncogenic miRNAs	miR-125b	DownregulatedUpregulated	SNAIL-1, SEMA4C STARD13	In vitro In vitro & in vivo	[65,66,67]
	miR-155	DownregulatedUpregulated	TCF4, ZEB2 C/EBPß, ZNF652	In vitro & in vivo In vitro & in vivo	[68,69,70,71]
	miR-200	DownregulatedUpregulated	ZEB1, ZEB2, FHOD1, PPM1F ZEB2, SEC23a, CDH1	In vitro In vitro & in vivo	[72,73,74,75,76,77,78]

**Table 3 cancers-13-00337-t003:** miRNAs regulating the Warburg effect.

	miRNA	Expression in Breast Cancer	Target(s)	Experimental Design	Reference(s)
Oncogenic miRNAs	miR-155	Upregulated	PIK3R1, FOXO3a, STAT3, C/EBPβ	In vitro & in vivo	[95,96,97]
	miR-27b	Upregulated	PDHX	In vitro	[98]
	miR-378(*) ^#^	Upregulated	ERRγ and GABPA	In vitro	[99]

^#^ miR-378(*) may also be known as miR-378a-5p [100].

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
