# Peer review of "Tiny miRNAs Play a Big Role in the Treatment of Breast Cancer Metastasis"

_cancers, 2021, doi:10.3390/cancers13020337_

Round 1

Reviewer 1 Report

  1. miRNA could also modulate breast cancer metastasis by other mechanisms such as affecting cancer stem cell properties, cross-talk between miRNA with other RNAs, miRNA transferred by exosomes. The authors may add a few paragraphs about these. 
  2. Is it necessary to describe every single siRNA listed in each table? The authors may summarize basic mechanisms (e.g., affected signaling pathways) about how miRNA modulate cancer cell metastasis in sections 2.1-2.4.
  3. It is better to mention whether the studies are in vitro or in vivo in Tables 1-3. 

Author Response

Thank you for your valuable comments! We have made the following amendments to the review paper based on your suggestions.

1. miRNA could also modulate breast cancer metastasis by other mechanisms such as affecting cancer stem cell properties, cross-talk between miRNA with other RNAs, miRNA transferred by exosomes. The authors may add a few paragraphs about these.

As we wish to focus mainly on how miRNA affects various stages in metastasis, we have chosen to incorporate this suggestion into various junctures of the paper. An introduction to the cross-talk between miRNA with other lncRNAs has been included in lines 59-64. An additional paragraph with a brief discussion on exosomal transfer of miRNAs has been included in lines 115-126, while cancer stem cells has been incorporated into the section 2.2 in lines 174-178.

2. Is it necessary to describe every single siRNA listed in each table? The authors may summarize basic mechanisms (e.g., affected signaling pathways) about how miRNA modulate cancer cell metastasis in sections 2.1-2.4.

A minor change in line 146 under section 2.1 has been made. As the affected signalling pathways vary for each miRNA, we have chosen to present studies surrounding the effects of each miRNA together.

3. It is better to mention whether the studies are in vitro or in vivo in Tables 1-3. 

The appropriate amendments have been made, with the inclusion of an additional column on ‘Experimental Design’ for Tables 1-3.

Reviewer 2 Report

Authors have performed interesting review on the role of miRNAs in induction of breast cancer metastasis and processes preceding its formation. In the second part they summarize the existing procedures of miRNA-based therapies with all the limitations that are connected with the mentioned therapies. The study is well designed and executed. The writing is largely clear and concise in good English style. I have no specific comments. Perhaps all the abbreviations should be explained when used for the first time.

Author Response

Thanks a lot for your nice words and kind help!

Authors have performed interesting review on the role of miRNAs in induction of breast cancer metastasis and processes preceding its formation. In the second part they summarize the existing procedures of miRNA-based therapies with all the limitations that are connected with the mentioned therapies. The study is well designed and executed. The writing is largely clear and concise in good English style. I have no specific comments. Perhaps all the abbreviations should be explained when used for the first time.

Response: The key abbreviations used have been explained, while proteins, cytokines and other factors are explained in the list of abbreviations at the end for better clarity of the overall text.

Reviewer 3 Report

MicroRNAs (miRNAs) are key regulators of metastasis in breast cancer. In this review manuscript, Teo et al. summarized important roles of miRNAs in breast cancer metastasis including the potentiation of angiogenesis, epithelial-mesenchymal transition, the Warburg effect, and the tumor microenvironment. The authors also summarized the recent development in miRNA-based therapies in breast cancer. In general, the manuscript is well-written, and it provides new information regarding the miRNAs in regulating breast cancer metastasis and progression. Here are my minor comments.

  1. Previous studies indicates that miRNAs are less abundant in tumors, including breast cancer, than in their normal tissue counterparts, leading to the notion that miRNAs are predominantly tumor suppressors rather than tumor promoters. Additionally, the authors also mentioned that “miRNAs have been implicated in each stage of cancer metastasis, acting either as tumour suppressors or oncogenic miRNAs to suppress or promote metastasis respectively (Line 104-105)” and “ miR-200 family is a double-edged sword (Line 229).” It is better to delete or change the term “oncogenic” in the sentences “…. the key roles of oncogenic miRNAs in breast cancer” as well as “ the roles of oncogenic miRNAs in four key areas of ……”

In the Figure 1, the authors should make sure that these are all oncogenic miRNAs. Otherwise, they also should change the term “oncogeneic.”

  1. Again, authors might add more information/discuss about tumor suppressor miRNAs in breast cancer in this review.

Author Response

MicroRNAs (miRNAs) are key regulators of metastasis in breast cancer. In this review manuscript, Teo et al. summarized important roles of miRNAs in breast cancer metastasis including the potentiation of angiogenesis, epithelial-mesenchymal transition, the Warburg effect, and the tumor microenvironment. The authors also summarized the recent development in miRNA-based therapies in breast cancer. In general, the manuscript is well-written, and it provides new information regarding the miRNAs in regulating breast cancer metastasis and progression. Here are my minor comments.

  1. Previous studies indicates that miRNAs are less abundant in tumors, including breast cancer, than in their normal tissue counterparts, leading to the notion that miRNAs are predominantly tumor suppressors rather than tumor promoters. Additionally, the authors also mentioned that “miRNAs have been implicated in each stage of cancer metastasis, acting either as tumour suppressors or oncogenic miRNAs to suppress or promote metastasis respectively (Line 104-105)” and “ miR-200 family is a double-edged sword (Line 229).” It is better to delete or change the term “oncogenic” in the sentences “…. the key roles of oncogenic miRNAs in breast cancer” as well as “ the roles of oncogenic miRNAs in four key areas of ……”

In the Figure 1, the authors should make sure that these are all oncogenic miRNAs. Otherwise, they also should change the term “oncogeneic.”

  1. Again, authors might add more information/discuss about tumor suppressor miRNAs in breast cancer in this review.

Response: Thank you for your nice comments and suggestions!

The relevant amendments have been made in lines 30, 73 and 125. We hoped to keep the bulk of discussion of section 2 on miRNAs involved in potentiating breast cancer metastasis, thus we chose to feature miRNAs that demonstrated oncogenic properties in breast cancer. However, the discussion was expanded to include miRNAs with either oncogenic or tumour suppressor miRNAs for miRNA-based therapies in breast cancer, as we believe (i) the suppression of oncogenic miRNA levels, or (ii) the increase of tumour suppressor miRNA levels are key steps which may be used as potential therapies against breast cancer.